# A Convenient U-Shape Microreactor for Continuous Flow Biocatalysis with Enzyme-Coated Magnetic Nanoparticles-Lipase-Catalyzed Enantiomer Selective Acylation of 4-(Morpholin-4-yl)butan-2-ol

Ali O. Imarah [1,2], Fausto M. W. G. Silva [1], László Tuba [1], Ágnes Malta-Lakó [1], József Szemes [1], Evelin Sánta-Bell [1] and László Poppe [1,3,4,*]

[1] Department of Organic Chemistry and Technology, Budapest University of Technology and Economics, Műegyetem rkp. 3, H-1111 Budapest, Hungary

[2] Chemical Engineering Department, College of Engineering, University of Babylon, Hilla Babylon 5100, Iraq

[3] Biocatalysis and Biotransformation Research Center, Faculty of Chemistry and Chemical Engineering, Babeș-Bolyai University of Cluj-Napoca, Arany János Str. 11, RO-400028 Cluj-Napoca, Romania

[4] SynBiocat Ltd., Szilasliget u 3, H-1172 Budapest, Hungary

* Correspondence: poppe.laszlo@vbk.bme.hu; Tel.: +36(1)463-3299

**Abstract:** This study implements a convenient microreactor for biocatalysis with enzymes immobilized on magnetic nanoparticles (MNPs). The enzyme immobilized onto MNPs by adsorption or by covalent bonds was lipase B from *Candida antarctica* (CaLB). The MNPs for adsorption were obtained by covering the magnetite core with a silica shell and later with hexadecyltrimethoxysilane, while for covalent immobilization, the silica-covered MNPs were functionalized by a layer forming from mixtures of hexadecyl- and 3-(2-aminoethylamino)propyldimethoxymethylsilanes in 16:1 molar ratio, which was further activated with neopentyl glycol diglycidyl ether (NGDE). The resulting CaLB-MNPs were tested in a convenient continuous flow system, created by 3D printing to hold six adjustable permanent magnets beneath a polytetrafluoroethylene tube (PTFE) to anchor the MNP biocatalyst inside the tube reactor. The anchored CaLB-MNPs formed reaction chambers in the tube for passing the fluid through and above the MNP biocatalysts, thus increasing the mixing during the fluid flow and resulting in enhanced activity of CaLB on MNPs. The enantiomer selective acylation of 4-(morpholin-4-yl)butan-2-ol (±)-**1**, being the chiral alcohol constituent of the mucolytic drug Fedrilate, was carried out by CaLB-MNPs in the U-shape reactor. The CaLB-MNPs in the U-shape reactor were compared in batch reactions to the lyophilized CaLB and to the CaLB-MNPs using the same reaction composition, and the same amounts of CaLB showed similar or higher activity in flow mode and superior activity as compared to the lyophilized powder form. The U-shape permanent magnet design represents a general and easy-to-access implementation of MNP-based flow microreactors, being useful for many biotransformations and reducing costly and time-consuming downstream processes.

**Keywords:** magnetic nanoparticles; enzyme immobilization; lipase; flow biocatalysis; reactor design; magnetic agitation; kinetic resolution; chiral morpholine derivative

## 1. Introduction

Day by day, biocatalysts become more and more important for the bio, food, fuel, and pharmaceutical industries [1–4].

Lipases (EC 3.1.1.3) are enzymes that metabolize fats by hydrolysis and are present in almost all the Earth's flora and wildlife. In biotechnological applications, lipases, mostly of bacteria, fungi, and yeasts, have gained increasing attention [5–7]. Lipases are utilized to make diverse substances such as octyl acetate, methyl salicylate, ethyl acetate, and ethyl lactate, among other short-chain esters [8]. They can also be used as an alternative approach for producing poly(butylene succinate) (PBS) [9] or trimethylolpropane triesters [10].

Cold active lipases are gaining popularity in the detergent industry. They also use less energy because they work at low temperatures [11]. Many biotechnological processes are accelerated at high temperatures. Therefore, thermostable lipases are the focus of significant research [12]. From a psychrophilic basidomycetous yeast *Pseudozyma antarctica* (first isolated from a lake in Antarctica, previously also named *Candida antarctica*), two lipases were isolated, purified, and named as lipases A and B (CaLA and CaLB, respectively) [13]. They were later cloned and overexpressed in the fungus *Aspergillus oryzae*. Nowadays, CaLA and CaLB, of microbial origin, are known for their extreme properties, rendering them useful biocatalysts for various applications [14,15]. In this study, the lipase B from *Candida antarctica* (CaLB) was selected as biocatalyst, due to the versatility and stability of this enzyme [16,17].

Enzyme immobilization can solve the problem of enzyme recovery and reuse, and may be used to improve enzyme stability, activity, selectivity, specificity, reducing inhibitions, and may be coupled with purification [18]. The major modes of immobilization are binding to carrier (by various modes such as adsorption by physical, ionic forces, covalent, or affinity binding), carrier-free methods (such as cross-linked enzyme aggregates (CLEAs or cross-linked enzyme crystals (CLECs)), or physical entrapment (within microcapsules or porous polymer matrices). However, not all immobilization protocols should be expected to produce all these positive results [19].

Immobilized lipases are effective biocatalysts for enzymatic synthesis [17–21] or for biosensing [22]. Immobilization of lipases on hydrophobic supports often results in activity enhancement by interfacial activation [21]. The most widely used immobilization of CaLB applies adsorption on a macroporous polymeric carrier [17]. Covalent immobilization of CaLB on functionalized macroporous polymeric carriers could result in thermal stabilization [23]. Entrapment of CaLB within sol–gel matrices could also enhance thermal and other properties of the biocatalyst [24]. Such a form of CaLB could be applied to synthesize a variety of taste esters for foods applied in a green and sustainable way [25].

With increased scientific understanding and nanotechnological innovation, the use of nano-sized support matrices for enzyme-based biotransformation bioprocesses is rising. Several nanomaterials are used as supports for enzyme immobilization for enhancing enzyme catalytic properties in industrial applications such as carbon nanotubes, graphene/graphene oxide, electrospun nanofibers, metal–organic frameworks, magnetic and non-magnetic nanoparticles, silica nanoparticles, and other nanohybrid matrices [26]. Accordingly, lipases immobilized on nano-carriers as robust biocatalysts have shown promise in recent years [5–7,27,28]. Electrospun fibrous systems proved to be advantageous forms of CaLB in enantiomer selective biotransformations [29] or in enzymatic Knoevenagel condensations [30]. Cross-linked enzyme adhered nanoparticles (CLEANs), a form of CaLB, were applied to produce bio-grade fragrance esters [31]. CaLB could be covalently attached to functionalized single-walled carbon nanotubes, creating an effective nano bioconjugate for kinetic resolution of various racemic 1-arylethan-1-ols in batch and flow modes [32].

Iron oxide nanoparticles are particles with a diameter of about 1 to 500 nm. They are also called magnetic nanoparticles (MNPs) and are used as a support in enzyme immobilization. Magnetite ($Fe_3O_4$) and maghemite ($\gamma$-$Fe_2O_3$) are the two most common types of superparamagnetic iron oxide. They have sparked broad interest as enzyme carriers due to their unique features [33–35]. Some of the superior features of these materials are the high surface area, low mass transfer resistance, and ease of enzyme isolation from the reaction mixture relative to other supports used for enzyme immobilization [33,36]. Lipases immobilized on heterogeneous magnetic carriers are easy to recover in biotransformations by magnetic field, thereby lowering operational costs and enhancing the purity of the products [35,37].

Heterofunctional supports proved to be advantageous for lipase immobilizations [38]. *Candida antarctica* lipase B (CaLB) has been also immobilized on hetero-functional MNPs by numerous methods, such as (i) adsorption on aminopropyltrimethoxysilane (ApTMOS)-modified MNPs for kinetic resolution (KR) of secondary alcohols [39], on Lys-modified

MNPs [40], on gallic acid-formaldehyde grafted MNPs [41], (ii) affinity binding of His-tagged CaLB on magnetite MNPs containing long-armed nickel-nitrilotriacetic acid surface groups [42], or by (iii) covalent binding using diazonium chemistry for immobilization on carbon-coated MNPs [43], epoxy functionalized polymer-grafted MNPs [44], or (after glutaraldehyde activation) onto chitosan-coated MNPs [45,46], onto ApTMOS-activated MNPs [47–50], onto Lys-modified MNPs [40], and (iv) a combination of covalent attachment and polymer embedding onto MNPs [51]. The above-mentioned CaLB-MNP biocatalysts were already applied to produce short- and medium-chain fatty esters, employed as taste components [52], flavor esters [50,51,53], biodiesel [54], or to perform kinetic resolution of racemic secondary alcohols [39,46]. It was also shown that ultrasound agitation improves CaLB-MNPs activity [52].

In addition to their use in shake flasks in batch mode, enzyme-coated MNPs could be used as biocatalysts in various reactor configurations [36,55], such as the chip-sized flow-through reactor with cells containing magnetically anchored MNP biocatalysts in bioreaction screening applications [56,57]. The performance of the MNP-based magnetic systems depends on the behavior of the MNPs under the influence of the magnetic field, which is related to the magnetic force, Stokes drag, and diffusive motions [58]. Mass transfer resistance at the boundary surface of a particle plays a crucial role in heterogeneous sorption from strongly diluted solutions, as it may often limit the total reaction rate [59]. It is therefore of particular interest to accelerate mass transfer in such systems. In the field of biocatalysis, microreactor technology is increasingly being used because of its cost-effectiveness and environmental sustainability [60,61]. Enzyme-coated MNP-based biocatalysts that have a large surface area are also applicable in various systems with decreased mass transfer limitations [34,55].

A major benefit of the MNP-based microreactors is the ease of replacing the MNP biocatalyst with fresh or other types of MNP biocatalysts, thereby creating a unique opportunity to develop modular micro-systems with the ability to have flexible variation of biocatalysts [56,57]. Numerous magnetic particle (MP)-based reactors were already developed. For chemical processes, a continuously stirred tank reactor (CSTR) setup was developed, using magnetic cation exchanger stirring with alternating current [59]. A tube reactor with external agitation with permanent magnets was applied for chemical kinetic resolution performed by MNP-bound chiral catalyst [62]. Biocatalytic processes were also performed in MNP-based reactors. For example, in a tubular device, cholesterol determination was performed with co-immobilized cholesterol esterase/cholesterol oxidase–MNP anchored with a permanent magnet [63]. A much higher biocatalytic efficiency was obtained with MNP-bound pectinase in an MNP-based membrane reactor than in a batch reactor [64]. Lipase-catalyzed methanolysis of castor oil was performed in a tube reactor containing electromagnet-agitated magnetic biocatalyst [65]. Lipase-catalyzed kinetic resolution of an amine was performed in a tube reactor, fluidizing co-immobilized magnetite-enzyme in cross-linked chitosan particles with an external electromagnet [66]. A multicell MagneChip device with permanent magnets for continuous flow biocatalysis was developed for biotransformations with MNP-bound phenylalanine ammonia-lyase [56,57]. It is worth noting that electromagnets are easier to control, but permanent magnets can generate a much stronger electric field.

Structural analysis of the marketed drugs revealed the phenyl ring as the most abundant ring fragment. The aromatic ring contains only $sp^3$ hybridized heavy atoms that are planar, thus the phenyl ring is the most typical 2D fragment. Following the most abundant 2D ring, the other three most frequently occurring ring fragments are morpholine, piperidine, and piperazine [67]. These saturated heterocycles are so-called 3D fragments containing at least one $sp^3$ hybridized heavy atom, rendering them non-planar. A later structural analysis of the marketed drugs indicated that "escaping from Flatland" due to 3D fragments—allowing more complex and less rigid molecules by introducing asymmetric centers—contributed to enhanced drug-likeness of the molecules without significantly increasing molecular weight [68]. Accordingly, morpholine is a critical structural component

of bioactive compounds and experimental medicines commonly employed in medicinal chemistry and pharmacology [69]. This scaffold and its derivative's frequent use in these fields are related to their favorable physicochemical, biological, and metabolic features and simple synthesis methods. Appropriately substituted morpholines have long been known to have many biological effects, including analgesic, anti-inflammatory, antioxidant activity, antimicrobial, and anticancer properties [69,70]. The two compounds with significant biological activities in Figure 1 illustrate bioactive molecules containing the morpholine ring. For example, Fedrilate is a centrally acting cough suppressant which was patented as a mucolytic by UCB [71,72]. Additionally, in vitro antitumor activity tests showed $IC_{50}$ value of 1.2 μM for compound 193 on Nagoya University-Gastric Cancer-3 (NUGC-3) cell line [73]. Since the alcohol part of these bioactive compounds, the 4-(morpholin-4-yl)butan-2-ol (**1**) moiety, is chiral, but was never synthesized in enantiopure form, the CaLB-MNP-catalyzed kinetic resolution of this heterocycle-containing scaffold has been selected as a major focus of this study.

**Figure 1.** Structure of the cough suppressant Fedrilate and Compound 193 with antitumor activity both containing the 4-(morpholin-4-yl)butan-2-ol (**1**) unit as common chiral element (shown in blue and marking the center of asymmetry by the symbol *).

Based on the above background, this study aimed to investigate new surface modification of MNPs for CaLB immobilization by adsorption and covalent binding and to apply the CaLB-MNPs in a convenient novel microreactor system to produce enantiomers of 4-(morpholin-4-yl)butan-2-ol (**1**) by acylation-based kinetic resolution.

## 2. Results and Discussion

### 2.1. Immobilization of Lipase B from Candida antarctica onto Magnetic Nanoparticles

As the CaLB-MNP biocatalyst for this study, lipase immobilization onto hydrophobic MNPs using simple adsorption and covalent binding was investigated (Scheme 1). The hydrophobic MNPs for adsorption were created by covering the magnetite core with a simple silica shell by tetraethoxysilane (TEOS) treatment, followed by etching with hexadecyltrimethoxysilane (HdTMOS). The heterofunctionalized MNPs for covalent binding were created by modification and a combination of our methods for bisepoxide activation of silica shell coated and aminopropyltrimethoxysilane (ApTMOS)-treated MNPs [74] and coating of silica nanoparticles [31] or silica-shelled MNPs [75] with mixed functions. Based on our previous experiences [75], the ratio of 1:16 for 3-(2-aminoethylamino)propyldimethoxymethylsilane (ApDMOMS) and HdTMOS was selected for heterofunctionalization prior to bisepoxide activation. Another study on tailoring the spacer arm for covalent immobilization of CaLB onto bisepoxide-activated aminoalkyl resins [23] indicated neopentyl glycol diglycidyl ether (NGDE) as a suitable spacer unit for activation of the aminoalkyl functions.

After creating the hexadecyl etched MNPs for adsorption ($MNP_A$) and the heterofunctionalized and NGDE-activated MNPs for covalent binding ($MNP_C$), the capacity of the MNP carriers for CaLB immobilization was investigated by applying CaLB:MNP mass ratio between 1:5 and 1:40. Thus, first, the immobilization yield as a function of CaLB loading and immobilization time was investigated (Figure 2).

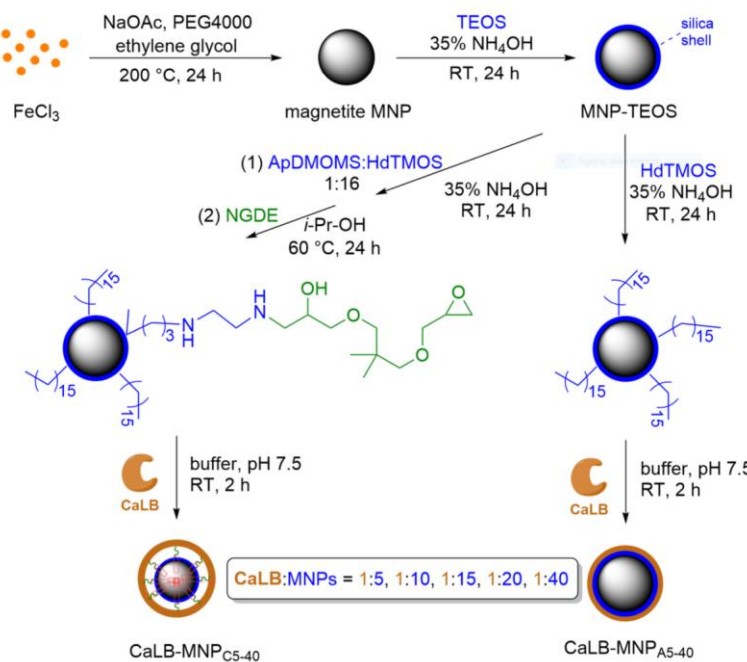

**Scheme 1.** Production of MNP carriers and immobilization of lipase B from *Candida antarctica* (CaLB) by adsorption (CaLB-MNP$_{An}$) or by covalent binding (CaLB-MNP$_{Cm}$) at room temperature (RT) (where n = 5–40 and m = 5–40; reflecting to the MNP:CaLB mass ratio applied).

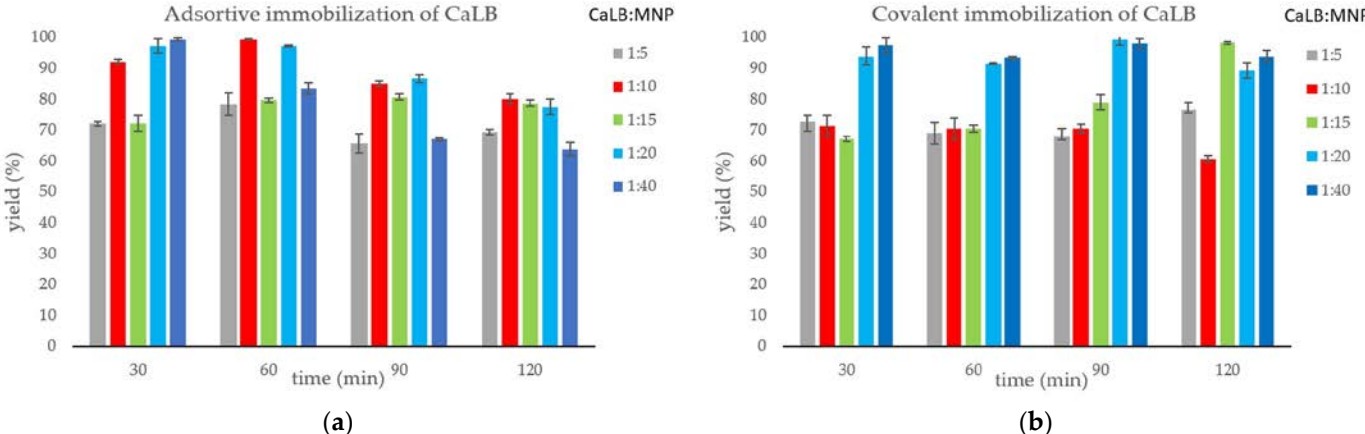

**Figure 2.** Immobilization yields for lipase B from *Candida antarctica* (CaLB) on MNPs at a different CaLB:MNP ratio (between 1:5 and 1:40) after various immobilization times (5 mg mL$^{-1}$ MNP carrier, CaLB (8.0, 4.0, 2.66, 2.0, 1.0 mg mL$^{-1}$), in sodium phosphate buffer (pH 7.5, 100 mM) at room temperature). The differently colored bars in the two charts show data of experiments (**a**) for adsorption immobilization, with MNPs etched by hexadecyl functions (HdTMOS) resulting in CaLB-MNP$_{An}$, and (**b**) for covalent binding onto MNPs etched by ApDMOMS:HdTMOS (1:16) and further activated with NGDE, resulting in CaLB-MNP$_{Cm}$ (where n = 5–40 and m = 5–40 reflect to the CaLB:MNP mass ratio applied).

As expected, the adsorption on the hexadecyl etched MNPs (MNP$_A$) proved to be rapid, as the highest immobilization yields ($Y_I$) were observed after the first 30 min period (Figure 2a). The lower immobilization yields after 60, 90, and 120 min in the adsorption experiments indicated a slower partial desorption of the fixed proteins following the rapid adsorption. Thus, out of the preparations immobilized by adsorption, CaLB-MNP$_{A10}$ (after 60 min immobilization time), which had a relatively high amount of CaLB

($90\ mg_{CaLB}\ g_{MNPa}^{-1}$) and was almost fully immobilized ($Y_I$ = 99%), was selected for the further experiments.

The covalent binding of CaLB on the heterofunctionalized and NGDE-activated MNPs (MNP$_C$) exhibited a somewhat lower binding capacity, but no desorption (Figure 2b). Since the preparation with CaLB:MNP ratio of 1:15 exhibited after 120 min almost full fixed the CaLB ($Y_I$ = 98%) in relatively high amounts ($61\ mg_{CaLB}\ g_{MNPc}^{-1}$), this biocatalyst was selected out of the covalently fixed forms for the further parts of the study.

### 2.2. Kinetic Resolution of Racemic 4-(Morpholin-4-yl)butan-2-ol (±)-1 with CaLB-MNP Biocatalysts in Batch Mode

After testing the immobilization yields, the two selected CaLB forms (CaLB-MNP$_{A10}$ and CaLB-MNP$_{C15}$) were investigated further in the desired kinetic resolution (KR) of racemic 4-(morpholin-4-yl)butan-2-ol (±)-1 using vinyl acetate (Scheme 2).

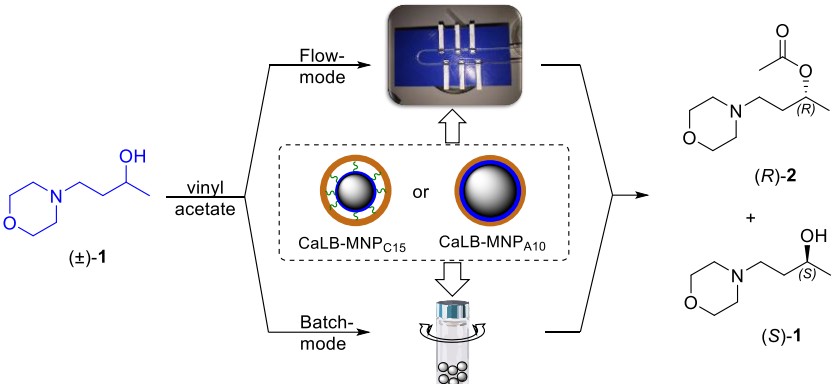

**Scheme 2.** Kinetic resolution of racemic 4-(morpholin-4-yl)butan-2-ol (±)-**1** with lipase B from *Candida antarctica* (CaLB) immobilized on magnetic nanoparticles (MNPs) in various reaction modes.

First, the time course of the KR process for racemic 4-(morpholin-4-yl)butan-2-ol (±)-**1** with the two selected CaLB biocatalysts was investigated in batch mode (Figure 3).

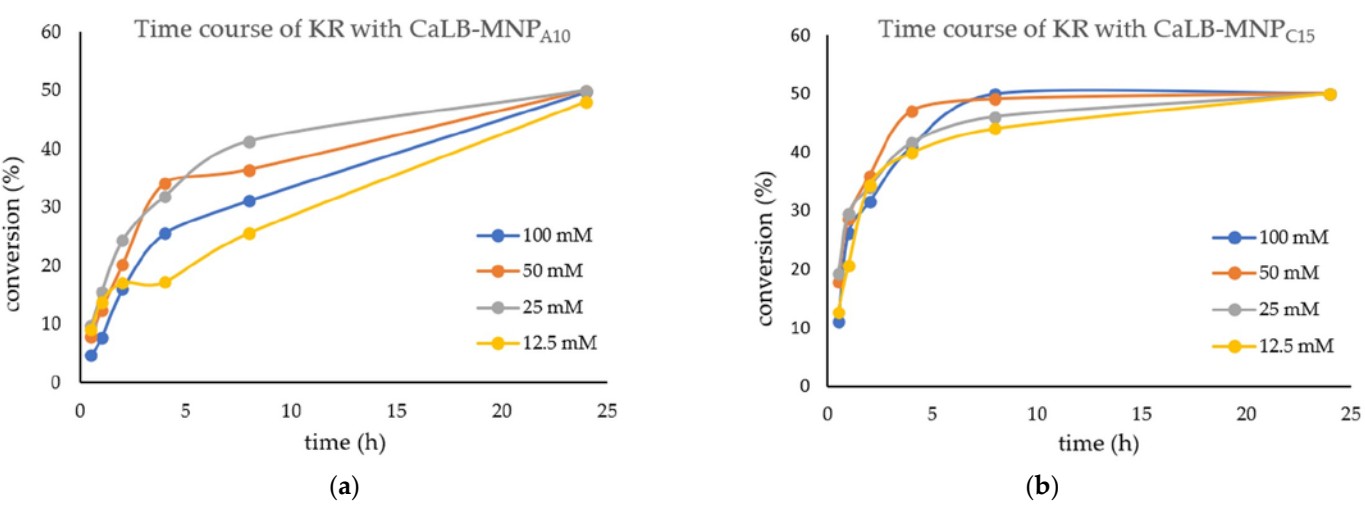

(**a**)　　　　　　　　　　　　　　　　　　　　　　　(**b**)

**Figure 3.** Time course of the kinetic resolution of racemic 4-(morpholin-4-yl)butan-2-ol (±)-**1** with vinyl acetate in batch mode used as a biocatalyst, (**a**) adsorptively immobilized CaLB-MNP$_{A10}$, or (**b**) covalently immobilized CaLB-MNP$_{C15}$ (for details see Section 3.3).

It is apparent that the kinetic resolutions with the aid of CaLB-MNP$_{C15}$ obtained by covalent attachment (Figure 3b) were faster at any substrate concentrations between 12.5–100 mM than those catalyzed by the CaLB-MNP$_{A10}$ prepared by adsorption immobi-

lization (Figure 3a), although CaLB-MNP$_{C15}$ contained a smaller amount of immobilized CaLB than the same amount of CaLB-MNP$_{A10}$. In fact, with CaLB-MNP$_{A10}$, the fastest process was observed at 25 mM (±)-**1** concentration (reaching 40% conversion after 8 h and the theoretically possible 50% conversion limit of a highly selective KR after 24 h) and the KRs with smaller or higher substrate concentrations were slower (Figure 3a). If the apparent $K_M$ of the faster reacting alcohol (R)-**1** is close to 10 mM, the slower rate of the KR at 12.5 mM concentration of (±)-**1** can be rationalized. At elevated concentrations of (±)-**1** (such as 50 and 100 mM), the enzyme is at saturation and can only transform a constant molar amount of (R)-**1**, meaning lower conversion at higher substrate concentrations.

The specific activity of CaLB for the acylation of racemic 4-(morpholin-4-yl)butan-2-ol (±)-**1** ($U_{CaLB}$) could be estimated at the low conversion regime ($c < 20\%$) where the linearity conditions are valid. Comparing the specific activities for (±)-**1** with the two kinds of biocatalysts in the reaction at 25 mM after 30 min revealed $U_{CaLB} = 90$ µmol min$^{-1}$ g$^{-1}$ for the CaLB-MNP$_{A10}$ form, while $U_{CaLB} = 261$ µmol min$^{-1}$ g$^{-1}$ for the CaLB-MNP$_{C15}$ form. Usually, the adsorptive immobilization results in higher specific activity of the biocatalyst than the covalent technique. We can rationalize our results by assuming that hydrophobic adsorption happens mostly at the hydrophobic site, forming by lid opening of the lipase and causing some steric hindrance for substrate access. In our case, however, the NGDE-activated carrier can fix the CaLB molecules at their surface-exposed nucleophilic residues (mostly at lysine residues), resulting in a higher proportion of fixing without steric hindrance at the active site entrance. The significant biocatalytic enhancement effect of the immobilization of CaLB was indicated by comparing the activity of CaLB-MNP forms to the lyophilized native form of the enzyme as well. In fact, by the same amount of CaLB powder as attached to the CaLB:MNPs at 1:10 mass ratio, only 1.4% conversion was observed after 24 h in the KR of (±)-**1** (meaning $U_{CaLB} < 1$ µmol min$^{-1}$ g$^{-1}$). This result, being a consequence of the significant mass transfer resistance within the nanopores of the micron-sized aggregate particles of the lyophilized form, indicated the importance of forming a monolayer of CaLB on the MNPs with high accessible surface area to eliminate these mass transfer issues.

Based on these experiments, the preparative scale KR of racemic 4-(morpholin-4-yl)butan-2-ol (±)-**1** was performed with the aid of CaLB-MNP$_{C15}$ at 25 mM substrate concentration (Table 1).

**Table 1.** Kinetic resolution of racemic 4-(morpholin-4-yl)butan-2-ol (±)-**1** with CaLB-MNP$_{C15}$ in batch mode.

| Product | Yield [1] (%) | ee [2] (%) | [α] [3] |
|---|---|---|---|
| (R)-**2** | 47.5 | >99 | +1.2 |
| (S)-**1** | 37.5 | >99 | +3.2 |
| (S)-**2** [4] | 90 [4] | >99 | −1.2 |

[1] From a kinetic resolution of (±)-**1** in batch mode after reaching 50% conversion ((±)-**1** (320 mg), CaLB-MNP$_{C15}$ (20 mg) and vinyl acetate (500 µL) in MTBE-hexane mixture (1:2 ratio, 10 mL) at room temperature (25 °C) for 24 h). [2] Determined by chiral GC. [3] Determined at the D-line of sodium, at 25 °C (c = 3, in ethanol). [4] Prepared by acetylation of (S)-**1** by AcCl/5M NaOH in ethyl acetate (see Section 4.4 of Supplementary Materials).

The enzymatic KR, at 320 mg substrate/20 mg CaLB-MNP$_{C15}$ scale, resulted in virtually enantiopure (R)-acetate (R)-**2** and (S)-alcohol (S)-**1** at room temperature (25 °C) in 24 h. The absolute configuration of the product was assumed by analogy with the (R)-configuration of the faster-reacting enantiomer of the isosteric 4-phenybutan-2-ol [76], and was confirmed by our established calculation method [77]. Chemical acetylation of the residual (S)-**1** with $[\alpha]_D^{25} = +3.2$ supported absolute configuration assignment, because the resulting (S)-acetate (S)-**2** had the same but opposite sign of optical rotation ($[\alpha]_D^{25} = -1.2$) as the (R)-acetate (R)-**2** produced by the CaLB-catalysis ($[\alpha]_D^{25} = +1.2$) (Table 1).

### 2.3. Kinetic Resolution of Racemic 4-(Morpholin-4-yl)butan-2-ol (±)-**1** with CaLB-MNP Biocatalysts in the U-Shape Reactor in Continuous Flow Mode

Although a MagneChip reactor design with six chamber anchoring MNP biocatalysts by properly positioned adjustable permanent magnets was already developed and tested for various biotransformations with phenylalanine ammonia-lyase [56,57], this reactor design required sophisticated manufacturing of the PDMS-based chip-like reactor body and allowed only low capacity charging with MNP biocatalyst (<1 mg per chamber).

Therefore, to perform the required enantiomer selective acylation of racemic 4-(morpholin-4-yl)butan-2-ol (±)-**1** with MNP-based CaLB biocatalyst in continuous flow mode on a larger scale, a new microreactor design was applied (Figure 4). The simple design applied a PTFE tube (ID = 0.75 mm) both as connection and as the body of reactor. The tube was placed in U-shape arrangement into a holder comprising six adjustable bars with permanent magnets. Positioning the permanent magnets under the tube could anchor the MNP-based catalysts within the tube, thus forming six reaction chambers in a tubular continuous flow reactor (Figure 4).

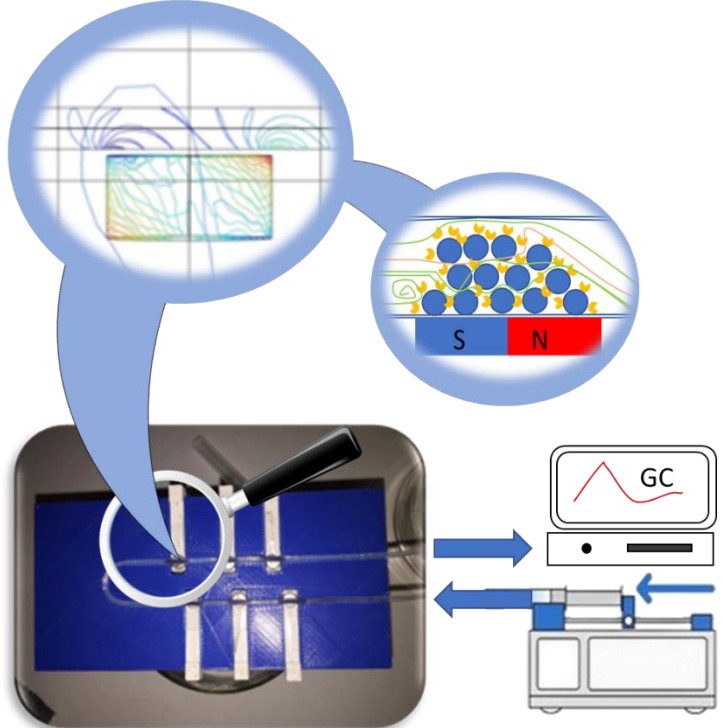

**Figure 4.** A simple U-shape MNP microreactor holding six adjustable permanent magnets under the PTFE tube reactor body. The MNP tube microreactor holder was created by 3D printing. The first excerpt shows the modeled magnetic field distribution within the empty tube. The second excerpt illustrates an idealized packing of the magnetically retained CaLB-MNPs.

Based on preliminary investigation in the U-shape reactor comprising PTFE tube of 0.75 mm ID charged with 5 mg of CaLB-MNP$_{C15}$ distributed in six reaction chamber parts at various flow rates (1, 2, and 3 μL min$^{-1}$; solution of (±)-**1** (25 mM) and vinyl acetate (2.5 equiv.) in MTBE-hexane 1:2 at room temperature (25 °C)), the 1 μL min$^{-1}$ flow rate was studied further.

First, the conditions for reaching the steady state within the U-shape reactor at various substrate concentrations (12.5–100 mM of (±)-**1**) were investigated (Figure 5a). In accordance with the substrate concentration dependence study of the KRs in batch mode (Figure 3b), the best results were achieved at 25 mM concentration. With 25 mM of (±)-**1**, the conversion reached 48%, being quite close to the theoretical limit of a fully selective KR,

within 8 h; at the lower (12.5 mM) or higher concentrations (50, 100 mM), the conversion within 8 h remained well below 40%. In a second run at 25 mM concentration using a preconditioned reactor (Figure 5b), the 40% conversion was exceeded even after 4 h, and the steady-state of a full KR (>48–49%) stabilized after 12 h and remained stable until 48 h.

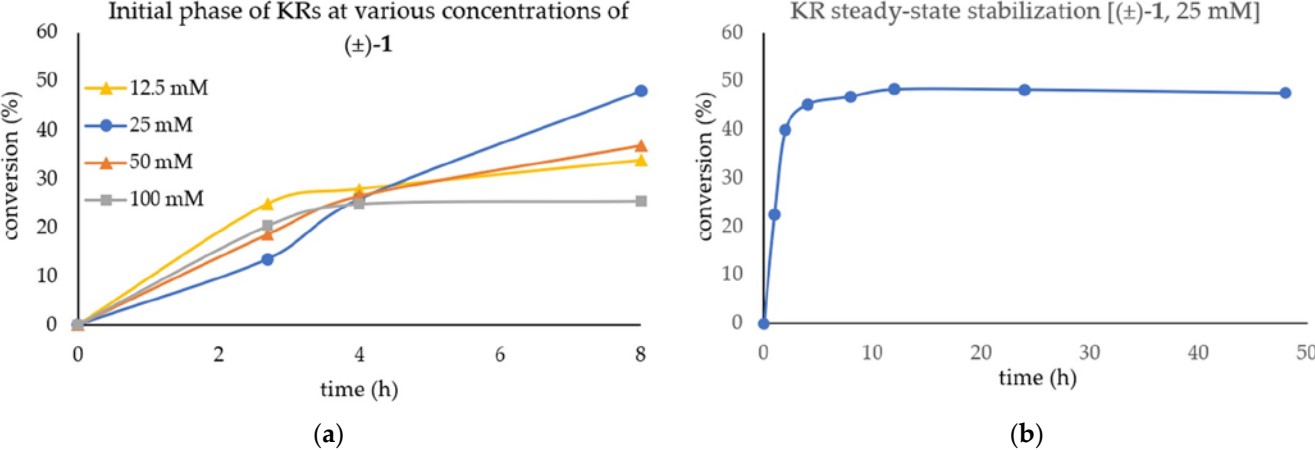

(**a**) (**b**)

**Figure 5.** Use of the U-shape reactor charged with CaLB-MNP$_{C15}$ (5 mg) for continuous flow mode kinetic resolution of racemic 4-(morpholin-4-yl)butan-2-ol (±)-**1** ((±)-**1** (varied concentrations), vinyl acetate (2.5 equiv.), MTBE:hexane 1:2, 25 °C, 1 μL min$^{-1}$). The (**a**) initial stabilization phase at various concentrations of (±)-**1** (12.5–100 mM) and (**b**) the time course of steady-state stabilization at 25 mM substrate concentration in the U-shape reactor are shown. The CaLB-MNP$_{C15}$ charged reactor used in (**a**) the concentration dependence investigations was recovered and applied in (**b**) the steady-state stabilization study.

To investigate the capability of the U-shape CaLB-MNP$_{C15}$ reactor system for preparative scale purposes and gain information on its long-term stability under continuous flow conditions, the preconditioned CaLB-MNP$_{C15}$ charged reactor for the preliminary flow reaction investigations was operated for a further 2 week period (Figure 6).

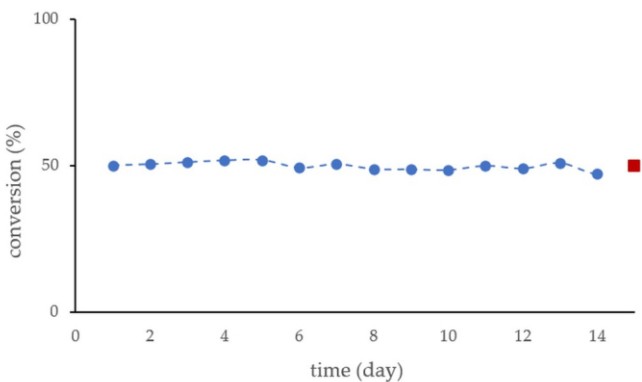

**Figure 6.** Long-term operational stability of the KR of racemic 4-(morpholin-4-yl)butan-2-ol (±)-**1** within the preconditioned U-shape CaLB-MNP$_{C15}$ reactor. The blue dots represent the conversion (50 ± 0.9%) in a sample at each day of a two weeks long run at room temperature (25 °C), while the red square shows the average. In each sample, the *ee* of products ((*R*)-**2** and (*S*)-**1**) exceeded 98%.

During the long term run of the U-shape CaLB-MNP$_{C15}$ reactor, 20 mL reaction mixture was collected. The average and standard error data ($c$ = 49.8 ± 1.33) for the samples during the steady state of the long-term run (Figure 6) were in good agreement with the conversion value ($c$ = 49.8%) determined for the collected total outflow solution. The products isolated from this mixture ((*R*)-**2**: Y = 48.0%, $[\alpha]^{25}_D$ = +1.2; (*S*)-**1**: Y = 40.5%, $[\alpha]^{25}_D$ = +3.2; *ee* > 99.5%

by GC for both) were indistinguishable from those which were isolated from the KR of (±)-**1** in batch mode.

Finally, the space time yields, starting from 25 mM (±)-**1** in the KR processes, could be compared. The KR process in batch mode revealed $STY_B = 4.0$ mmol $L^{-1}$ $h^{-1}$ for the (*R*)-acetate ((*R*)-**2**) formation. When the total U-shape volume ($V_{U\text{-shape}}$ = 118 μL) for the continuous flow KR process, starting from 25 mM (±)-**1**, was considered, a $STY_{U\text{-shape}}$ = 1.8 mmol $L^{-1}$ $h^{-1}$ could be determined. However, considering only the volume of the MNP-filled reaction chambers ($V_{U\text{-shape}}$ = 28 μL) of the U-shape MNP reactor, a $STY_{filled}$ = 7.4 mmol $L^{-1}$ $h^{-1}$ could be calculated. The situation $STY_{filled}$ being two times higher than $STY_B$ resembled to comparing a continuous flow KR in a packed-bed reactor to a KR batch mode [78].

## 3. Materials and Methods

### 3.1. Materials

CaLB for immobilization experiments (recombinant *Candida antarctica* lipase B produced by microbial fermentation in *Pichia pastoris*, exhibiting a single band around 33 kDa on SDS-gel EF; provided as lyophilized powder, Lot-NO: MA-b-0002, activity: 59,900 TBU/g) was obtained from c-LEcta (Leipzig, Germany).

Details on materials, solvents, methods for CaLB immobilization, parameters necessary to define the immobilized enzyme preparation [79], and synthesis of the racemic 4-(morpholin-4-yl)butan-2-ol (±)-**1** substrate are given in Supplementary information.

### 3.2. Analytical and Separation Methods

Optical rotations were measured on a Perkin-Elmer 241 polarimeter (Waltham, MA, USA) at the D-line of sodium in ethanol. The polarimeter was calibrated with measurements of both enantiomers of menthol.

Gas chromatographic (GC) analyses were performed with an Agilent 4890 gas chromatograph (Santa Clara, CA, USA) equipped with FID detector using $H_2$ carrier gas (injector: 250 °C, detector: 250 °C, head pressure: 12 psi, split ratio: 50:1) using Hydrodex β-6TBDM column (25 m × 0.25 mm × 0.25 μm film of heptakis-(2,3-di-*O*-methyl-6-*O*-*t*-butyldimethylsilyl)- β-cyclodextrin; Macherey and Nagel, Düren, Germany). Details on methods and retention times of components in the kinetic resolution reactions are given in Table S1 in Supplementary information.

### 3.3. Comparative Activity Tests of the CaLB-MNP Biocatalysts by Acetylation of (±)-**1**

The biocatalytic activities of the various CaLB-MNP biocatalysts were assayed and compared using the enzymic acylation reactions of the racemic 4-(morpholin-4-yl)butan-2-ol (±)-**1** with vinyl acetate as acylating agent.

The assays for the various CaLB-MNP preparations were performed in clean screw-capped vials (1.5 mL, 8-425 Vial Small Opening V813/V817, Aijiren Technology Inc., Zhejiang, China), using CaLB-MNPs (5 mg) as biocatalyst in the reaction mixture containing racemic 4-(morpholin-4-yl)butan-2-ol (±)-**1** (12.5, 25, 50, or 100 mM) and vinyl acetate (31.5, 62.5, 125, or 250 mM, respectively) in a mixture of methyl *t*-butyl ether (MTBE)-hexane (1:2 ratio, 1 mL) at room temperature (25 °C) in a Vibramax 100 shaker (Heidolph, Schwabach, Germany).

For sampling from reactions under various conditions, samples (20 μL) taken from the reaction mixtures were diluted with ethanol (480 μL) and analyzed by GC as described in Section 3.2. The preliminary activity test in batch mode performed as triplicates indicated a variation within 2%. Therefore, further optimization and time course experiments were performed as single series.

### 3.4. Kinetic Resolution of (±)-**1** by the CaLB-MNP_{C15} Biocatalysts in Batch Mode

Into a 20 ml screw-cap vial, racemic 4-(morpholin-4-yl)butan-2-ol (±)-**1** (320 mg, 2 mmol), CaLB-MNP_{C15} (50 mg), and vinyl acetate (500 μL) were added to a mixture of methyl *t*-butyl ether (MTBE)-hexane (1:2 ratio, 10 mL), and the suspension was shaken at room temperature

(25 °C) in a Vibramax 100 shaker for 24 h. The CaLB-MNP$_{C15}$ biocatalyst was removed by anchoring with the aid of a neodymium magnet and decanting. The resuspended CaLB-MNP$_{C15}$ was washed by a mixture of methyl *t*-butyl ether (MTBE)-hexane (1:2 ratio, 2 × 2.5 mL). After removing the volatiles from the combined solutions by vacuum rotary evaporation, the resulted products were separated by chromatography on a silica gel column (silica gel (15 g), dichloromethane-methanol-35% ammonia solution 10:1:0.5 as eluant) to give the formed enantiopure (*R*)-acetate (*R*)-**2** and residual (*S*)-alcohol (*S*)-**1**.

### 3.4.1. (*R*)-4-(Morpholin-4-yl)butan-2-yl acetate (*R*)-**2**

Yield: 191 mg, 47.5%. $[\alpha]_D^{25}$ = +1.2 (*c* 3, EtOH) for the sample having *ee* > 99.5% by GC. $[\alpha]^{25}_D$ = −1.2 (*c* 3, EtOH) for the sample having *ee* > 99.5% by GC. IR (film, cm$^1$): 2955, 2853, 2810, 1734, 1459, 1448, 1372, 1242, 1118, 1071, 1013, 867. $^1$H NMR (500 MHz, CDCl$_3$, δ ppm) 1.25 (d, *J* = 6.2 Hz, 3H, -C*H*$_3$), 1.70 (m, 1H, CH*H*-CH-O), 1.82 (m, 1H, C*H*H-CH-O), 2.03 (s, 3H, -COC*H*$_3$), 2.39 (m, 4H, CHCH$_2$C*H*$_2$), 2.45 (br s, 4H, NC*H*$_2$CH$_2$O), 3.73 (t, *J* = 4.6 Hz, 4H, NCH$_2$C*H*$_2$O), 4.97 (m, 1H, C*H*-O). $^{13}$C NMR (126 MHz, CDCl$_3$, δ ppm): 20.09, 21.33, 32.50, 53.54, 55.02, 66.56, 69.37, 170.66.

### 3.4.2. (*S*)-4-(Morpholin-4-yl)butan-2-ol (*S*)-**1**

Yield: 119 mg, 37.5%. $[\alpha]_D^{25}$ = +3.2 (*c* 3, EtOH) for the sample having *ee* > 99.5% by GC. IR (film, cm$^1$): 3389, 2961, 2852, 2614, 1453, 1448, 1288, 1115, 1031, 912, 867, 769. $^1$H NMR (300 MHz, CDCl$_3$, δ ppm) 1.15 (*dd*, 3H, -C*H*$_3$), 1.50 (*m*, 1H, CH*H*-CH-O), 1.63 (*m*, 1H, C*H*H-CH-O), 2.41 (*br s*, 2H, CHCH$_2$C*H*$_2$), 2.62 (*m*, 4H, NC*H*$_2$CH$_2$O), 3.70 (*br s*, 4H, NCH$_2$C*H*$_2$O), 3.94 (*m*, 1H, C*H*-O), 4.9 (*br s*, 1H, C-O*H*). $^{13}$C NMR (75 MHz, CDCl$_3$, δ ppm): 22.5, 33.0, 53.7, 58.1, 66.8, 69.6.

### 3.5. Design and Assembly of the U-Shape MNP Reactor

The U-shape MNP reactor system (Figure 4) was designed by the AutoCAD (2020 student version) program and printed by a Rankfor100 3D printer (CEI Conrad Electronic International, Ltd., New Territories, Hong Kong). For this study, neodymium disc magnets, 4 mm × 2 mm, N35 (Euromagnet Ltd, Budapest, Hungary), were as permanent magnets, and polytetrafluoroethylene (PTFE) tube ID 0.75 mm was used as the reactor body and for connection parts.

### 3.6. Kinetic Resolution of (±)-**1** by the CaLB-MNP$_{C15}$ Biocatalysts in the U-Shape Reactor in Continuous Flow Mode

The U-shape reactor for continuous flow kinetic resolution comprised a PTFE tube (ID = 0.75 mm) placed in a holder, allowing the adjustable positioning of six permanent magnets under the tube. This formed six positions at which CaLB-MNP$_{C15}$ could be anchored by the magnets within the tube (Figure 4).

To be comparable to the results of the KR in batch mode, similar reaction conditions were applied for the KR in the U-shape reactor (CaLB-MNP$_{C15}$ amount (5 mg), concentration of substrate (±)-**1** (25 mM), and vinyl acetate concentration (62.5 mM) in a mixture of methyl *t*-butyl ether (MTBE)-hexane (1:2 ratio), room temperature (25 °C)) as applied in the KRs in batch mode (see Section 3.3).

For filling the six chambers of the reactor, suspensions of CaLB-MNP$_{C15}$ (6 × 0.83 mg MNPs suspended in 1 mL of solvent at 50 µL min$^{-1}$) were fed in a counter-current direction to the later fluid flow, starting from the last chamber until the first one.

Experiments were performed at various flow rates (1, 2, 3 µL min$^{-1}$) to achieve different residence times (8, 4, 2.5 min, respectively). Sampling was performed in a similar way, as described in Section 3.3.

For the preparative scale experiment, the U-shape MNP tube reactor was fed with racemic 4-(morpholin-4-yl)butan-2-ol (±)-**1** (40 mg, 0.25 mmol) and vinyl acetate (231 µL, 2.5 equiv.) in 10 mL of solvent (MTBE–hexane 1:2) at a flow rate of 1 µL min$^{-1}$. After collecting the outflowing solutions (20 mL, 14 days), the volatiles were evaporated by

vacuum rotary evaporation, and the resulting products were separated by chromatography on a silica gel column (silica gel (5 g), dichloromethane-methanol-35% ammonia solution 10:1:0.5 as eluant) to leave the formed enantiopure (*R*)-acetate (*R*)-**2** and (*S*)-alcohol (*S*)-**1**.

### 3.6.1. (*R*)-4-(Morpholin-4-yl)butan-2-yl acetate (*R*)-**2**

Yield: 38.4 mg, 48.0%. $[\alpha]^{25}_D$ = +1.2 (*c* 3, EtOH) for a sample having *ee* > 99.5% by GC.

### 3.6.2. (*S*)-4-(Morpholin-4-yl)butan-2-ol (*S*)-**1**

Yield: 32.4 mg, 40.5%. $[\alpha]^{25}_D$ = +3.2 (*c* 3, EtOH) for a sample having *ee* > 99.5% by GC.

## 4. Conclusions

In this study, a convenient U-shape tubular microreactor for MNP-catalysis was developed and applied for kinetic resolution of the pharmaceutically relevant racemic 4-(morpholin-4-yl)butan-2-ol (±)-**1** with lipase B from *Candida antarctica* immobilized on magnetic nanoparticles. In the new U-shape MNP reactor, the enzyme-coated MNPs, forming nanochannels within the individual MNPs, allowed passing the fluid through and above the MNP biocatalysts, thus breaking the streamline of laminar flow and increasing the mixing during the fluid flow. The mixing of the streamline for fluid flow enabled enhanced effective activity of enzymes. The results of a KR process with a covalently immobilized CaLB-MNP biocatalyst (CaLB-MNP $_{C15}$) in the U-shape reactor operated under continuous flow conditions compared to the KR with the same reaction composition and the same amount of CaLB-MNP $_{C15}$ in batch mode confirmed this effective enhancement effect.

The U-shape tubular microreactor developed for this study represents a general and easy-to-access implementation of the MNP-based flow microreactor technology for various biotransformations, avoiding costly and time-consuming manufacturing and downstream processes.

**Supplementary Materials:** The following are available online at https://www.mdpi.com/article/10.3390/catal12091065/s1, Materials and methods for preparation of MNP carriers and CaLB immobilization; Synthesis of (±)-**1**, and the racemic acetate and (*S*)-acetate ((±)-**2** and (*S*)-**2**, respectively); Table S1: GC method and retention times for analysis of the kinetic resolution reactions by chiral GC; Figure S1–S17: $^1$H NMR spectrum of 4-(morpholin-4-yl)butan-2-one; and GC chromatograms, $^1$H- and $^{13}$C-NMR, and IR spectra of compounds (±)-**1**, (±)-**2**, (*R*)-**2**, and (*S*)-**1** [76,77,80–82].

**Author Contributions:** Conceptualization, L.P. and A.O.I.; methodology, A.O.I., L.T., Á.M.-L., E.S.-B. and L.P.; validation, L.P. and A.O.I.; investigation, A.O.I., F.M.W.G.S., L.T. and J.S.; resources, L.P.; writing—original draft preparation, A.O.I., F.M.W.G.S. and Á.M.-L.; writing—review and editing, A.O.I. and L.P.; visualization, A.O.I. and L.P.; supervision, L.P.; funding acquisition, L.P. All authors have read and agreed to the published version of the manuscript.

**Funding:** The research reported in this paper is part of project no. TKP2021-EGA-02, implemented with the support provided by the Ministry for Innovation and Technology of Hungary from the National Research, Development and Innovation Fund, financed under the TKP2021 funding scheme. The National Research, Development and Innovation Office (Budapest, Hungary) is acknowledged for funding (SNN-125637). This work was also supported by a grant of the Romanian Ministry of Education and Research, CCCDI-UEFISCDI, project number PN-III-P2-2.1-PED-2019-5031, within PNCDI III.

**Data Availability Statement:** Not applicable.

**Acknowledgments:** Technical assistance of Edit Tóth Németh in lab work is thankfully acknowledged.

**Conflicts of Interest:** The authors declare no conflict of interest.

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
