# Peer review of "A Convenient U-Shape Microreactor for Continuous Flow Biocatalysis with Enzyme-Coated Magnetic Nanoparticles-Lipase-Catalyzed Enantiomer Selective Acylation of 4-(Morpholin-4-yl)butan-2-ol"

_catalysts, doi:10.3390/catal12091065_

Round 1

Reviewer 1 Report

The presented technique for CaLB immobilization and its use in a magnetic-field assisted microreactor is a nice example of the implementation of enzymes in industrially relevant kinetic resolution reaction aiming to harness the advantages of miniaturized reactors.

However, the paper is not well structured and has to be rewritten before further consideration. Besides, several superficialities need to be addressed.

Comments and suggestions for improvement:

1.   The introduction is too long (3 pages) and covers too broad a literature survey (66 references). Extensive basic on lipases and a very well-known CaLB and their use is not necessary for the audience of the Catalysis journal. Also, a whole-page intro on MNPs and their use as immobilization supports could be shortened and more focused. On the other hand, there are almost no basics on the rationale for using magnetic-field assisted microreactor, which is a core topic of the research. The benefits of using microreactors with magnets and other examples of microreactor configurations with permanent or electromagnets used for biocatalytic reactions are not covered.

2.   Yeasts belong to fungi (line 41); maybe just point out the microbial origin of CaLB.

3.   Abbreviations should be first introduced with the whole word (e.g. KR in line 88, RT in Scheme 1).

4.   It is not clear to which ratio the authors are referring when speaking of CaLB:MNP ratio. Please specify.

5.   Figure 3 and Figure 5: the legend should be clarified in the figure caption.

6.   Figure 4 could be improved as it’s not very clear what is represented in the enlarged frames.

7.   Figure 6: What means the red symbol in the graph?

8.   There’s no statistical analysis of data.

Author Response

The presented technique for CaLB immobilization and its use in a magnetic-field assisted microreactor is a nice example of the implementation of enzymes in industrially relevant kinetic resolution reaction aiming to harness the advantages of miniaturized reactors.

However, the paper is not well structured and has to be rewritten before further consideration. Besides, several superficialities need to be addressed.

Comments and suggestions for improvement:

  1. The introduction is too long (3 pages) and covers too broad a literature survey (66 references). Extensive basic on lipases and a very well-known CaLB and their use is not necessary for the audience of the Catalysis journal. Also, a whole-page intro on MNPs and their use as immobilization supports could be shortened and more focused. On the other hand, there are almost no basics on the rationale for using magnetic-field assisted microreactor, which is a core topic of the research. The benefits of using microreactors with magnets and other examples of microreactor configurations with permanent or electromagnets used for biocatalytic reactions are not covered.

Response: According to your opinion lipases and even Lipase B from Candida antarctica (CaLB) and magnetic nanoparticles (MNPs) are well known for the readers of Catalysts and the introduction should be shortened to focus mostly on MNP-based microreactors. However, the Academic Editor and Reviewers 2 and 3 have contradictory opinion on this issue, as they even ask various extensions and citations on behavior of lipases, and general issues on immobilization.

Because our work has three novel aspects i) new way of immobilization of CaLB on MNPs, ii) a new setup for MNP-based microreactor for biocatalysis, and iii) a new kinetic resolution of a heterocyclic secondary alcohol of direct pharmaceutical interest, our Introduction tried to inform the readers on the background of all the three aspects. As conclusion of the common opinions of the reviewers we try to complete the Introduction with all the requested extensions but keeping the extensions as short as possible. According to your opinion, the Introduction of the revised MS is extended with mentioning the benefits of the MNP-based microreactors and other examples of microreactor configurations with permanent or electromagnets used for biocatalytic reactions, as follows:

“A major benefit of the MNP-based microreactors is the ease of replacing the MNP biocatalyst with fresh or even other type of MNP-biocatalyst, thereby creating a unique opportunity to develop modular micro-systems with the ability of flexible variation of biocatalysts [54,55]. Numerous magnetic-particles (MP)-based reactors were already developed. For chemical processes, a continuously stirred tank reactor (CSTR) setup was developed using magnetic cation exchanger stirring with alternating current [57]. A tube reactor with external agitation with permanent magnets was applied for chemical kinetic resolution performed by MNP-bound chiral catalyst [60]. Biocatalytic processes were also performed in MNP-based reactors. For example, in a tubular device cholesterol determination was performed with co-immobilized cholesterol esterase/cholesterol oxidase-MNP anchored with a permanent magnet [61]. A much higher biocatalytic efficiency was obtained with MNP-bound pectinase in an MNP-based membrane reactor than in a batch reactor [62]. Lipase-catalyzed methanolysis of castor oil was performed in tube reactor containing electromagnet-agitated magnetic biocatalyst [63]. Lipase-catalyzed kinetic resolution of an amine was performed in a tube reactor fluidizing co-immobilized magnetite-enzyme in cross-linked chitosan particles with an external electromagnet [64]. A multicell MagneChip device with permanent magnets for continuous-flow biocatalysis was developed for biotransformations with MNP-bound phenylalanine ammonia-lyase [54,55]. Worth noting that electromagnets are easier to control, but permanents magnets can generate much stronger electric field.”

However, the other parts are also extended according to the requests of the other Reviewers and the Academic Editor.

  1. Yeasts belong to fungi (line 41); maybe just point out the microbial origin of CaLB.

Response: This issue has been solved in the next sentence as shown: “Nowadays CaLA and CaLB of microbial origin are known for their extreme properties rendering them useful biocatalysts for various applications …”

  1. Abbreviations should be first introduced with the whole word (e.g. KR in line 88, RT in Scheme 1).

Response: These issues have been solved as detailed below.

Line 88: “… for kinetic resolution (KR) of secondary alcohols …”;

Legend to Scheme 1: “Production of MNP carriers and immobilization of lipase B from Candida antarctica (CaLB) by adsorption (CaLB-MNPAn) or by covalent binding (CaLB-MNPCm) at room temperature (RT) [where n= 5-40 and m= 5-40; reflecting to the CaLB:MNP mass ratio applied].”

  1. It is not clear to which ratio the authors are referring when speaking of CaLB:MNP ratio. Please specify.

Response: CaLB:MNP mass ratio were applied. This issue is corrected at any instances like in the extended legend for Scheme 1. Moreover, for the best preparations, the CaLB(mg) per MNP(g) values are also given.

  1. Figure 3 and Figure 5: the legend should be clarified in the figure caption.

Response: Figure captions have been added to Figures 3 and 5 as requested.

  1. Figure 4 could be improved as it’s not very clear what is represented in the enlarged frames.

Response: The legend of Figure 4 is extended as follows: “… The first excerpt shows the modeled magnetic field distribution within the empty tube. The second excerpt illustrates an idealized packing of the magnetically retained CaLB-MNPs.”

  1. Figure 6: What means the red symbol in the graph?

Response: The Legend to Figure 5 is extended to clarify the meaning of the colored dots and the red symbol. “… The blue dots represent the conversion (50±0.9%) in a sample at each day of a two weeks long run at room temperature (25 °C), while the red square shows the average. …”

  1. There’s no statistical analysis of data.

Response: Error bars for the measurements in triplicate have been added to Figure 2. Moreover, a sentence has been added to the revised MS: “The average and standard error data (c= 49.8±1.33) for the samples during the steady state of the long-term run (Figure 6) were in good agreement with the conversion value (c= 49.8%) determined for the collected total outflow solution.”

Reviewer 2 Report

1. In this study, dose authors compared the activity of enzyme after immobilization and I which form it was maximum (free, adsorb and covalent). it should be mention in the manuscript.

2. Author did not mention error bar in figure 2, it should be mention in all figure of this study. Authors must have to mention how many time the same experiment was performed.

3. Why authors did not perform the HPLC for study the separation of racemic mixture?

4. In material method part did not mention the source of lipase producing strain. It should be mention.

5. Author also did not mention any reference in Material and Method part, please try to add some latest reference as following

Microbial lipases and their industrial applications: a comprehensive review.

Surfactant-mediated permeabilization of Pseudomonas putida KT2440 and use of the immobilized permeabilized cells in biotransformation

Production, immobilization and characterization of beta-glucosidase for application in cellulose degradation from a novel Aspergillus versicolor

Lipases: sources, immobilization methods, and industrial applications

 Temperature-resistant and solvent-tolerant lipases as industrial biocatalysts: Biotechnological approaches and applications.

Immobilization of transaminase from Bacillus licheniformis on copper phosphate nanoflowers and its potential application in the kinetic resolution of RS-α-methyl benzyl amine

Author Response

Comments and Suggestions for Authors

  1. In this study, dose authors compared the activity of enzyme after immobilization and I which form it was maximum (free, adsorb and covalent). it should be mention in the manuscript.

Response: A new paragraph dealing with these issues was added to the Results and discussion section. “The specific activity of CaLB for the acylation of racemic 4-(morpholin-4-yl)butan-2-ol (±)-1 (UCaLB) could be estimated at the low conversion regime (c <20%) where the linearity conditions are valid. Comparing the specific activities for (±)-1 with the two kinds of biocatalysts in the reaction at 25 mM after 30 min revealed UCaLB= 90 µmol min-1 g-1 for the CaLB-MNPA10 form, while UCaLB= 261 µmol min-1 g-1 for the CaLB-MNPC15 form. Usually, the adsorptive immobilization results in higher specific activity of the biocatalyst than the covalent technique. We can rationalize our results by assuming that hydrophobic adsorption happens mostly at the hydrophobic site forming by lid opening of the lipase causing some steric hindrance for substrate access. In our case, however, the NGDE activated carrier can fix the CaLB molecules at their surface exposed nucleophilic residues (mostly at lysine residues) resulting in higher proportion of fixing without steric hindrance at the active site entrance. The significant biocatalytic enhancement effect of the immobilization of CaLB was indicated by comparing the activity of CaLB-MNP forms to the lyophilized native form of the enzyme as well. In fact, by the same amount of CaLB powder as attached to the CaLB:MNPs at 1:10 mass ratio, only 1.4% conversion was observed after 24 h in the KR of (±)-1 (meaning UCaLB < 1 µmol min-1 g-1). This result–being a consequence of the significant mass transfer resistance within the nanopores of the micron-sized aggregate particles of the lyophilized form–indicated the importance of forming a monolayer of CaLB on the MNPs with high accessible surface area to eliminate these mass transfer issues. ”

  1. Author did not mention error bar in figure 2, it should be mention in all figure of this study. Authors must have to mention how many time the same experiment was performed.

Response: In the sub-section 3.3. of Materials and methods it was stated “The preliminary activity test in batch mode performed as triplicates indicated a variation within 2%, therefore further optimization and time course experiments were performed as single series.”

Accordingly, error bars for the measurements in triplicate have been added to Figure 2. Moreover, a sentence has been added to the revised MS: “The average and standard error data (c= 49.8±1.33) for the samples during the steady state of the long-term run (Figure 6) were in good agreement with the conversion value (c= 49.8%) determined for the collected total outflow solution.”

  1. Why authors did not perform the HPLC for study the separation of racemic mixture?

Response: As mentioned in Section 3.2, gas chromatographic (GC) analyses were performed with an Agilent 4890 gas chromatograph equipped with FID detector using H2 carrier gas using Hydrodex β-6TBDM column [with film of heptakis-(2,3-di-O-methyl-6-O-t-butyldimethylsilyl)- β-cyclodextrin]. As the further details on method and retention times of components in the kinetic resolution reactions (Table S1 in Supplementary information) indicate, both the alcohol enantiomers ((S)-1: 14.18 min, (R)-1: 14.39 min), and the acetate enantiomers ((S)-2: 17.03 min, (R)-2: 17.55 min) were separable. Usually, the accuracy of a chiral GC measurement using FID detection exceeds the accuracy of a chiral HPLC using a DAAD detector.

  1. In material method part did not mention the source of lipase producing strain. It should be mention.

Response: The information on CaLB indicated in the Supplementary information is added to the MS Materials and Methods section as the first sentence of Section 3.1: “CaLB for immobilization experiments (recombinant Candida antarctica lipase B as lyophilized powder) was obtained from c-LEcta (Leipzig, Germany).”

  1. Author also did not mention any reference in Material and Method part, please try to add some latest reference as following
  • Microbial lipases and their industrial applications: a comprehensive review.
  • Surfactant-mediated permeabilization of Pseudomonas putida KT2440 and use of the immobilized permeabilized cells in biotransformation
  • Production, immobilization and characterization of beta-glucosidase for application in cellulose degradation from a novel Aspergillus versicolor
  • Lipases: sources, immobilization methods, and industrial applications
  • Temperature-resistant and solvent-tolerant lipases as industrial biocatalysts: Biotechnological approaches and applications.
  • Immobilization of transaminase from Bacillus licheniformis on copper phosphate nanoflowers and its potential application in the kinetic resolution of RS-α-methyl benzyl amine

Response: Two of the above mentioned reviews (Microbial lipases and their industrial applications: a comprehensive review; and Temperature-resistant and solvent-tolerant lipases as industrial biocatalysts) have been added to the citations in the Introduction.

Reviewer 3 Report

The present paper describes the enzymatic acylation of 4-(Morpho- 4 lin-4-yl)butan-2-ol in a microreactor under continuous mode. The paper is very interesting and represents some advancement over the actual state-of-the-art. However, some revisions are required before it could be considered for publication, as follows:

- Abstract: The better experimental results obtained in this study for immobilized protein concentration and selective acylation should be briefly described. It is more interesting to the readers.

- Figure 2: The authors should report in its legend the initial protein loading and experimental conditions used to perform this study. The effect of activation step with NGDE on the immobilization process should be better discussed. The introduction of such groups reduced the immobilization of the enzyme. Moreover, the authors could also discuss the effect of different CaLB:MNP ratio (recommend express initial protein loading in terms of mg per gram of support) on the catalytic activity and specific activity (ratio between catalytic activity and immobilized protein concentration). See Boudrant et al., 2019 – https://doi.org/10.1016/j.procbio.2019.11.026

This is very interesting to the readers. See some articles. Moreover, I recommend cite these studies:

https://doi.org/10.1016/j.ijbiomac.2020.07.021

https://doi.org/10.1016/j.cattod.2020.03.059

https://doi.org/10.1016/j.biombioe.2021.106302

- Results and discussion: All experimental data should be represented with error bars. Explain the number replications used in the study.

- Materials and methods: The methodology used to prepare the different heterogeneous biocatalysts via physical adsorption and physical adsorption/covalent attachment should be described in detail.

Author Response

Comments and Suggestions for Authors

The present paper describes the enzymatic acylation of 4-(morpholin-4-yl)butan-2-ol in a microreactor under continuous mode. The paper is very interesting and represents some advancement over the actual state-of-the-art. However, some revisions are required before it could be considered for publication, as follows:

  • Abstract: The better experimental results obtained in this study for immobilized protein concentration and selective acylation should be briefly described. It is more interesting to the readers.

Response: According to the opinion of the Reviewer, the following extended sentence has been added to the Abstract: “The CaLB-MNPs in the U-shape reactor were compared in batch reactions to the lyophilized CaLB and to the CaLB-MNPs using the same reaction composition and the same amounts of CaLB showing similar or higher activity in flow mode and superior activity as compared to the lyophilized powder form.”

  • Figure 2: The authors should report in its legend the initial protein loading and experimental conditions used to perform this study. The effect of activation step with NGDE on the immobilization process should be better discussed. The introduction of such groups reduced the immobilization of the enzyme. Moreover, the authors could also discuss the effect of different CaLB:MNP ratio (recommend express initial protein loading in terms of mg per gram of support) on the catalytic activity and specific activity (ratio between catalytic activity and immobilized protein concentration). See Boudrant et al., 2019 https://doi.org/10.1016/j.procbio.2019.11.026 This is very interesting to the readers.

Response: The legend to Figure 2 has been extended as follows: “Immobilization yields for lipase B from Candida antarctica (CaLB) on MNPs at different CaLB:MNP ratio (between 1:5 and 1:40) after various immobilization times [5 mg mL-1 MNP carrier, CaLB (8.0, 4.0, 2.66, 2.0, 1.0 mg mL-1), in sodium phosphate buffer (pH 7.5, 100 mM) at room temperature].”

The effect of activation step with NGDE on the immobilization process should is discussed as follows: “We can rationalize our results by assuming that hydrophobic adsorption happens mostly at the hydrophobic site forming by lid opening of the lipase causing some steric hindrance for substrate access. In our case, however, the NGDE activated carrier can fix the CaLB molecules at their surface exposed nucleophilic residues (mostly at lysine residues) resulting in higher proportion of fixing without steric hindrance at the acive site entrance. The significant biocatalytic enhancement effect of the immobilization of CaLB was indicated by comparing the activity of CaLB-MNP forms to the lyophilized native form of the enzyme as well. In fact, by the same amount of CaLB powder as attached to the CaLB:MNPs at 1:10 mass ratio, only 1.4% conversion was observed after 24 h in the KR of (±)-1 (meaning UCaLB < 1 µmol min-1 g-1). This result–being a consequence of the significant mass transfer resistance within the nanopores of the micron-sized aggregate particles of the lyophilized form–indicated the importance of forming a monolayer of CaLB on the MNPs with high accessible surface area to eliminate these mass transfer issues.”

The reference on parameters necessary to define an immobilized enzyme preparation has been added to the MS. Because mass ratio of CaLB to MNP (from 1:5 to 1:40) is used throughout the MS, we would use this indicator for keeping consistency but add the mg(enzyme) per g(support). The specific activity in the acylation of racemic 4-(morpholin-4-yl)butan-2-ol (±)-1 with the optimal variants is added to the Discussion as follows: “Comparing the specific activities for (±)-1 with the two kinds of biocatalysts in the reaction at 25 mM after 30 min revealed UCaLB= 90 µmol min-1 g-1 for the CaLB-MNPA10 form, while UCaLB= 261 µmol min-1 g-1 for the CaLB-MNPC15 form.”

The protein loading and immobilization yield data are given as follows: “… the CaLB-MNPA10 (after 60 min immobilization time) having relatively high amount of CaLB (90 mgCaLB g-1MNPa) being almost fully immobilized (YI= 99%) …“; “… the preparation with CaLB:MNP ratio of 1:15 exhibited after 120 min almost full fixing of the CaLB (YI= 98%) in relatively high amounts (61 mgCaLB g-1MNPc),  …”

-    See some articles. Moreover, I recommend cite these studies:

https://doi.org/10.1016/j.ijbiomac.2020.07.021

https://doi.org/10.1016/j.cattod.2020.03.059

https://doi.org/10.1016/j.biombioe.2021.106302

Response: The above-mentioned articles have been cited at proper places.

  • Results and discussion: All experimental data should be represented with error bars. Explain the number replications used in the study.

Response: Error bars for the measurements in triplicate have been added to Figure 2. Moreover, a sentence has been added to the revised MS: “The average and standard error data (c= 49.8±1.33) for the samples during the steady state of the long-term run (Figure 6) were in good agreement with the conversion value (c= 49.8%) determined for the collected total outflow solution.”

  • Materials and methods: The methodology used to prepare the different heterogeneous biocatalysts via physical adsorption and physical adsorption/covalent attachment should be described in detail.

Response: Since the major focus of the article is to use MNP-based CaLB catalyst for kinetic resolution of racemic 4-(morpholin-4-yl)butan-2-ol (±)-1 in the U-shape reactor (being in agreement with the opinion of Reviewer 1)  we would keep our decision to describe the details of the immobilization processes as additional data in Electronic Supplementary Information.

Round 2

Reviewer 3 Report

The corrections were made, as suggested. However, the authors should add some information on the lipase (powder lipase formulation - see lines 356/357) used in this study such as protein concentration, activity and specific activity. It is lyophilized powder a crude or purified lipase? After corrections, I recommend that the manuscript is accpeted for publication in this journal.

Author Response

The corrections were made, as suggested. However, the authors should add some information on the lipase (powder lipase formulation - see lines 356/357) used in this study such as protein concentration, activity and specific activity. It is lyophilized powder a crude or purified lipase? After corrections, I recommend that the manuscript is accpeted for publication in this journal.:

Response: According to the opinion of the Reviewer, the following extended sentence disclosing all information about the CaLB frohas been added to the Materials and methods section: “CaLB for immobilization experiments (recombinant Candida antarctica lipase B produced by microbial fermentation in Pichia pastoris, exhibiting a single band around 33 kDa on SDS-gel EF; provided as lyophilized powder, Lot-NO: MA-b-0002, activity: 59,900 TBU/g) was obtained from c-LEcta (Leipzig, Germany).”